# Not All Pairs are Equal: Hierarchical Learning for Average-Precision-Oriented Video Retrieval

## ABSTRACT

The rapid growth of online video resources has significantly promoted the development of video retrieval methods. As a standard evaluation metric for video retrieval, Average Precision (AP) assesses the overall rankings of relevant videos at the top list, making the predicted scores a reliable reference for users. However, recent video retrieval methods utilize pair-wise losses that treat all sample pairs equally, leading to an evident gap between the training objective and evaluation metric. To effectively bridge this gap, in this work, we aim to address two primary challenges: **a)** The current similarity measure and AP-based loss are suboptimal for video retrieval; **b)** The noticeable noise from frame-to-frame matching introduces ambiguity in estimating the AP loss. In response to these challenges, we propose the *Hierarchical learning framework for **A**verage-Precision-oriented **V**ideo **R**etrieval (**HAP-VR**)*. For the former challenge, we develop the TopK-Chamfer Similarity and QuadLinear-AP loss to measure and optimize video-level similarities in terms of AP. For the latter challenge, we suggest constraining the frame-level similarities to achieve an accurate AP loss estimation. Experimental results present that HAP-VR outperforms existing methods on several benchmark datasets, providing a feasible solution for video retrieval tasks and thus offering potential benefits for the multi-media application.

## CCS CONCEPTS

• **Information systems** → *Video search*; *Learning to rank*.

## KEYWORDS

Video Retrieval; Average Precision; Hierarchical Similarity Optimization; Self-supervised Learning

## 1 INTRODUCTION

The rapid expansion of online video resources has made content-based video retrieval a crucial component for multi-media applications such as recommendation, video editing, and online education [1, 25]. As a fundamental task, video retrieval, which aims to efficiently and effectively rank candidate videos based on their similarities to the query video, has raised a wave of studies in the multi-media community.

Recent video retrieval methods [29, 31–33, 52] employ neural network models to learn video similarities by aggregating fine-grained

**Unpublished working draft. Not for distribution.**

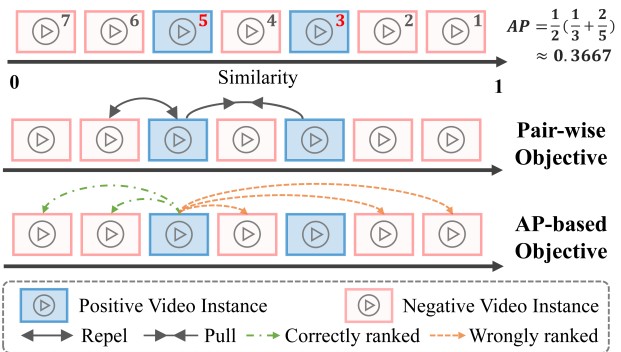

$$AP = \frac{1}{2}\left(\frac{1}{3} + \frac{2}{5}\right) \approx 0.3667$$

**Figure 1: Average Precision (AP) measures the average ranking of positive instances within a list, providing a comprehensive evaluation of the overall performance of the retrieval results. Pair-wise training objectives focus solely on pulling the positive instances closer while repelling the negative ones, failing to align with the AP metric. In contrast, AP-based objectives ensure this alignment by rectifying the rankings of mis-ranked positive-negative pairs in the list.**

embeddings of video frames, which have achieved a remarkable success compared with the early hand-craft approaches [11, 15, 27, 55, 58]. Nevertheless, **these models are typically optimized by pair-wise training objectives such as triplet loss, which are inconsistent with the evaluation metric.** Concretely, as a standard evaluation metric, Average Precision (AP) focuses on the top list by assigning larger weights to the top-ranked positive videos. Since the retrieval results are commonly processed sequentially in downstream tasks, the performance of the top list becomes crucial, thus making AP a more comprehensive metric as it better reflects this practical requirement. However, as shown in fig. 1, pair-wise losses treat all mis-ranked video pairs equally, ignoring the relative rankings among the instance list. This leads to an evident gap between the training objective and the evaluation metric, calling for an effective AP-based objective function to bridge this gap.

To solve a similar problem in image retrieval, a promising method is to optimize AP directly. Due to the non-differentiability of AP, existing methods concentrate on developing differential approximations of the AP [4, 6, 7, 46, 48, 57]. Although these AP optimization methods have achieved notable success in the image field, they cannot be directly applied to videos due to the complexity arising from the additional temporal dimension. Generally, the challenges are two-fold:

**a) The current similarity measure and the surrogate AP loss are suboptimal for AP-oriented video retrieval.** Typically, in mainstream frameworks, image similarities are measured with the cosine similarity of an embedding pair, while video similarities are aggregated from the redundant temporal-spatial features. Since

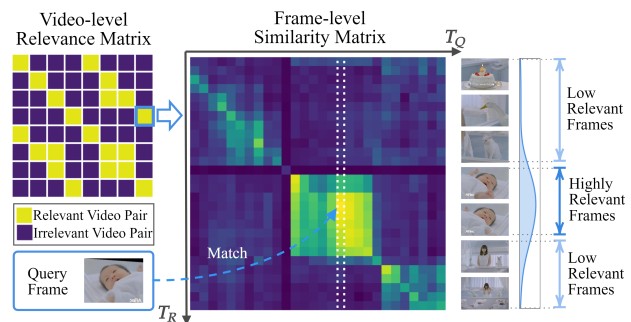

**Figure 2: Two relevant videos might not exhibit consistent relevance across all frame pairs due to the obvious redundancy and noise in the temporal dimension. Specifically, only a few consecutive frames in the candidate video are relevant to a given query frame.**

AP jointly considers the rankings of all instances rather than merely distinguishing whether a pair of videos matches, it is necessary to design a fine-grained similarity measure function for videos. Additionally, existing surrogate AP losses like Smooth-AP [4] suffer from a vanishing gradient when a sample pair is seriously misranked, leading to inefficient optimization. This phenomenon is more obvious for videos since the various video similarities fall into the gradient vanishing area more frequently.

**b) The noisy frame-to-frame matching leads to a biased AP estimation.** As illustrated in fig. 2, two relevant videos may not exhibit uniform relevance across all frames. Without fine-grained annotations, this ambiguity leads to false positive matching. In this case, the weights of the top-ranked videos might be reduced, hindering the AP loss from concentrating on the top list.

Based on the above considerations, we propose the *Hierarchical learning framework for Average-Precision-oriented Video Retrieval (HAP-VR)*, which contains video-level and frame-level constraints as detailed following:

To tackle challenge **a)**, we propose a topK-based similarity measure and a variant of AP loss with proper gradients. As a core component of our framework, the proposed *TopK-Chamfer Similarity* aggregates video-level similarities from frame-level similarities. Compared with previous maximum/average aggregations, the TopK-Chamfer Similarity retains fine-grained information while filtering out false correlations, providing refined video similarity for the following AP loss estimation. Another core component is a new surrogate AP loss, namely *QuadLinear-AP*, which enjoys a more reasonable distribution of gradients to rectify mis-ranked positive-negative pairs efficiently.

In search of a solution to challenge **b)**, we propose to correct the frame-level similarities without requiring fine-grained annotations. Motivated by the recent advance in self-supervised learning [9, 20, 24], we leverage the pre-trained vision model [8] to extract frame-level representations. Subsequently, we generate pseudo labels indicating the matched frames from the gap between these representations and distill the frame-level information to avoid ambiguity, leading to a more precise estimation of AP loss.

To summarize, the contributions of this work are three-fold:

- We develop a self-supervised hierarchical learning framework for Average-Precision-oriented video retrieval, named HAP-VR, to fill the gap between training objectives and evaluation metrics that the previous work has overlooked.
- Within HAP-VR, we propose the TopK-Chamfer Similarity and QuadLinear-AP loss to measure and optimize video-level similarities of the AP metric, alongside constraining frame-level similarities to produce a precise estimation of AP loss.
- Our experimental evaluation of HAP-VR across several large-scale benchmark datasets often presents a superior performance in terms of AP, ensuring its effectiveness for content-based video retrieval tasks.

## 2 RELATED WORK

In this section, we will introduce several previous works that contribute to video retrieval and Average Precision optimization.

### 2.1 Video Retrieval

Based on the granularity of similarity processing, video retrieval methods can be generally classified into two schemes, i.e. coarse-grained method and fine-grained method.

*2.1.1 Coarse-grained Method.* This kind of method focuses on extracting and aggregating features into a vector space, representing each video by a single vector to compute similarity at the video level. In the early stage, methods such as Bag-of-Words [5, 50], code books [30, 36] encode videos into a single vector by summarizing the extracted features through statistical aggregation, which neglect the temporal and spatial structures of the video. With the advent of deep learning in the video field, later approaches have started to train deep neural networks with metric learning [31, 35], promoting the transition from coarse-grained methods to fine-grained methods in the subsequent research.

*2.1.2 Fine-grained Method.* This kind of method typically extracts features from frames and thus generates multiple vectors to represent a video. Due to the utilization of a more enriched feature representation with spatial and temporal structures, fine-grained methods typically outperform coarse-grained methods within the same period. Early fine-grained methods focus on designing video temporal alignment solutions, e.g. temporal Hough Voting [15, 27], graph-based Temporal Network [55, 58] and Dynamic Programming [11], to match similar segments within the videos through hand-craft algorithms. Following the development of methods like TMK [44] and LAMV [2], which use Fourier transform and kernel tricks for spatio-temporal representation learning, there has been a shift towards transformer-based architectures [16, 22, 23, 44, 52]. Among these methods, TCA [52] adopts a self-attention mechanism to capture temporal relationships among fine-grained features and utilizes a contrastive learning strategy for training, VRL [23] combines CNN with a transformer structure to train a model without labels. Recent methods concentrate on designing neural networks to learn similarity functions for calculating video-level similarities from original video representations. ViSiL [29] provides a supervised learning method that designs a 4-layer CNN to obtain video-level similarities from frame-level similarities. Additionally,

Figure 3: The architecture of our proposed framework. The data batch is processed through a feature extractor to obtain patch-level embeddings. Afterward, we compute frame-level and video-level similarity matrices utilizing spatial and temporal correlation aggregation modules in sequence. Simultaneously, the batch is fed into a pre-trained self-supervised model to generate pseudo labels that indicate frame-level relevance. Ultimately, we apply the QuadLinear-AP to both the frame-level and video-level similarity matrices and backpropagate the loss to optimize the model's parameters.

DnS [33] employs knowledge distillation to train the student networks with ViSiL serving as a teacher network. More recently, $S^2VS$ [32] proposes a self-supervised learning approach built on the foundations of an improved structure of ViSiL. Despite previous studies developing increasingly complex models, their reliance on training with pair-wise objectives has led to a misalignment with the evaluation metric. Furthermore, these efforts typically focus solely on optimizing video-level similarity, neglecting the importance of frame-level similarity learning. In our work, we propose a method that employs an AP-based objective to bridge this gap, hierarchically optimizing AP for both frame-level and video-level similarities during the learning process.

## 2.2 Average Precision Optimization

Traditional metric learning methods provide a learning paradigm for retrieval tasks, mapping instances into an embedding space and employing distance metrics to design pair-wise objective functions such as contrastive loss [10] or triplet loss [59]. However, these methods merely focus on increasing the distance between positive and negative instances within pairs or tuples, neglecting to improve the overall ranking of positives more comprehensively. This narrow focus can lead to overfitting, particularly in the face of imbalanced data distribution. A promising method is ranking-based metric learning with AP as the target. However, the non-differentiability of ranking terms in AP poses a challenge, obstructing the update of model parameters during backpropagation. To address this issue, numerous AP optimization methods have been developed. Listwise

approaches [6, 7, 46, 57] utilize differentiable histogram binning to optimize loss functions based on ranking lists. Others provide structured learning frameworks based on SVM [39, 60] or conduct direct loss minimization [19, 53] to optimize AP. Moreover, Rolinek et al. [48] introduce BlackBox combinatorial solvers [43] to differentiate the ranking terms in AP. More recently, Brown et al. [4] propose Smooth-AP to use the Sigmoid function for approximating the indicator function, offering a simple and efficient way to differentiate AP. However, approximation methods like Smooth-AP neglect the gradient vanishing in the low AP area. To this end, we propose QuadLinear-AP, a novel loss for AP optimization, to designate appropriate gradients to the improperly ranked positive-negative pairs, ensuring the efficiency of the optimization process.

## 3 METHODOLOGY

### 3.1 Task Definition

In the video space $\mathcal{X}$, each video can be seen as a tensor $V = \{v_j \in \mathbb{R}^{H \times W \times C}\}_{j=1}^{T}$ where $T, H, W$, and $C$ represent the dimension of time, height, width, and channel, respectively. Given a pair of videos $V_1, V_2 \in \mathcal{X}$, video similarity learning aims to learn a similarity function $f : \mathcal{X} \times \mathcal{X} \to \mathbb{R}$, such that $f(V_1, V_2)$ represents the relevance between $V_1, V_2$. During the training stage, at each step, we sample a batch of videos $B = \{V_i \in \mathcal{X}\}_{i=1}^{N}$ where the length of $V_i$ is $T_i$. Let $Y \in \{0, 1\}^{N \times N}$ be the video-level relevance matrix, where $Y_{ij} = 1$ if $V_i$ and $V_j$ are relevant or $Y_{ij} = 0$ otherwise. For the sake of presentation, we denote the similarity score as $s_{ij} = f(V_i, V_j)$, and denote the rankings among positive/negative subsets as $S^{k+} =$

$\{s_{ki} = f(V_k, V_i)|V_i \in B, Y_{ki} = 1, k \neq i\}$, $S^{k-} = \{s_{ki} = f(V_k, V_i)|V_i \in B, Y_{ki} = 0\}$.

According to the above definition, we aim to optimize $f$ such that $f(V_k, V_i) > f(V_k, V_j)$ if $Y_{ki} = 1$ and $Y_{kj} = 0$, such that it achieves a higher AP score:

$$\max_f AP(f) = \frac{1}{N} \sum_{k=1}^{N} AP_k(f),$$

$$AP_k(f) = \frac{1}{|S^{k+}|} \sum_{s_{ki} \in S^{k+}} \frac{\mathcal{R}(s_{ki}, S^{k+})}{\mathcal{R}(s_{ki}, S^{k+} \cup S^{k-})}, \quad (1)$$

where $\mathcal{R}(s, S) = 1 + \sum_{s' \in S} \mathcal{H}(s' - s)$ is the descending ranking of $s$ in $S$, $\mathcal{H}(\cdot)$ is the Heaviside function [45], i.e., $\mathcal{H}(x) = 1$ if $x > 0$ otherwise $\mathcal{H}(x) = 0$.

## 3.2 Overview

We aim to design an AP-oriented framework for video similarity learning to align the training objective with the evaluation metric of video retrieval. As illustrated in fig. 3, given two videos $V, V' \in \mathcal{X}$, we first utilize a feature extractor $g(\cdot)$ to extract patch-level embeddings $g(V), g(V') \in \mathbb{R}^{T \times R \times D}$, where $T$, $R$, $D$ are the number of frames, patches, and the embedding dimension, respectively. Afterward, the patch-to-patch similarities are measured with the cosine similarity, resulting in a patch-level similarity matrix $S(V, V') \in \mathbb{R}^{T \times R \times R \times T'}$.

Next, we optimize the similarity measure in a hierarchical strategy. At the video level, we aggregate the spatial and temporal correlation to video-level similarities via the proposed TopK-Chamfer Similarity (detailed in section 3.3). Following ViSiL [29], we also apply a CNN to propagate the inter-frame similarities. Afterward, the video-level similarities are input into the proposed QuadLinear-AP loss. As outlined in section 3.4, for the frame-level constraint, we leverage a pre-trained vision model to generate pseudo labels and distill the frame-to-frame similarities to our feature extractor with the QuadLinear-AP loss.

## 3.3 Video-oriented AP Optimization

In this subsection, we first implement the similarity function $f$ through a bottom-up video similarity measure to map patch-level embeddings into similarities. Following this, we propose an AP surrogate loss with appropriate gradients for optimization, instructing $f$ to rank the similarities accurately.

*3.3.1 Bottom-up Video Similarity Measure.* In this subsection, we present the detailed process of feature aggregation. Specifically, given a pair of videos, we first aggregate the patch-level similarities $S(V, V') \in \mathbb{R}^{T \times R \times R \times T'}$ along the spatial dimension, leading to a frame-level similarity matrix $m_s(V, V') \in \mathbb{R}^{T \times T'}$. Afterward, we aggregate the temporal dimension as the video-level similarity $f(V, V') = m_t(V, V')$. Consider a batch of videos $B = \{V_i\}_{i=1}^{N}$, similarities of all pairs form an $N \times N$ video-level similarity matrix.

Early work utilizes a maximum/average operator to gather the fine-grained features. Kordopatis-Zilos et al. [29] suggest that two relevant frames/videos might be similar only in a part of region/period. From this perspective, to gather the spatial features, they propose to focus on the most similar region in $g(V')$ for each query patch in

$g(V)$, leading to the Chamfer-Similarity-based aggregation [3]:

$$m_s(V, V')_{x,y} = \frac{1}{R} \sum_{i=1}^{R} \max_{j=1,\cdots,R} S(V, V')_{x,i,j,y}. \quad (2)$$

The above operator identifies the maximum score for each query patch and averages these scores of all query patches in a frame to reflect the similarity between two frames. A similar operation is performed to gather the temporal features.

However, focusing on the maximum score makes the similarity measure sensitive to spatial noises caused by distractors. Besides, different from the patch-to-patch similarity matrix with a fixed shape, the temporal dimension in videos is flexible and varies greatly. Furthermore, given a query video $V_k$ and two relevant candidate videos $V_1, V_2$, the Chamfer Similarity might assign equal similarities for both $V_1$ and $V_2$, even if $V_2$ contains more relevant frames. Such a phenomenon reduces the distinguishability of positive samples, leading to an ambiguous ranking estimation.

Therefore, we seek a fine-grained similarity measure to estimate a precise AP loss. Specifically, rather than taking the maximum value, we jointly consider the top K scores:

$$m_s(V, V')_{x,y} = \frac{1}{RK} \sum_{i=1}^{R} \sum_{j=1}^{K} S(V, V')_{x,i,[j],y}, \quad (3)$$

where $K = k_s \times R$ and $S(V, V')_{x,i,[j],y}$ refers to the $j$-th largest value, or formally: $S(V, V')_{x,i,[1],y} \geq \cdots \geq S(V, V')_{x,i,[R],y}$.

On top of the frame-level similarities, following ViSiL [29], we utilize a CNN block $\psi$ to fuse the frame-to-frame similarities:

$$\bar{m}_s(V, V') = \psi\left(m_s(V, V')\right) \in \mathbb{R}^{\frac{T}{s} \times \frac{T}{s}}, \quad (4)$$

where $s > 1$ is the downsampling factor of $\psi$. In this way, the frame-level similarity is mapped into a learnable measure space. Additionally, it downscales the similarity matrix to reduce the computational burden. Afterward, we utilize the proposed TopK-Chamfer Similarity in the temporal dimension, leading to the video-level similarity:

$$f(V, V') = m_t(V, V') = \frac{1}{T} \sum_{i=1}^{T} \sum_{j=1}^{K} \bar{m}_s(V, V')_{i,[j]}, \quad (5)$$

where $K = k_t \times T'$ and $\bar{m}_s(V, V')_{i,[1]} \geq \cdots \geq \bar{m}_s(V, V')_{i,[T']}$. On one hand, compared with the original Chamfer Similarity, the TopK-Chamfer Similarity maintains fine-grained information; on the other hand, compared with the average operator, it avoids the disturbing noise introduced by the irrelevant frames.

*3.3.2 Gradient-Enhanced AP Surrogate Loss.* To effectively update the fine-grained similarity measure, in this part, we propose a new surrogate AP loss, such that it enjoys proper gradients in the mis-ranked area.

For a batch of videos $B = \{V_i \in \mathcal{X}\}_{i=1}^{N}$, recall that for a query video $V_k$, the similarity scores of the relevant and irrelevant videos are denoted as $S^{k+} = \{s_{ki} = f(V_k, V_i)|V_i \in B, Y_{ki} = 1, k \neq i\}$ and $S^{k-} = \{s_{ki} = f(V_k, V_i)|V_i \in B, Y_{ki} = 0\}$, respectively. For the sake of presentation, let $d_{ji}^k = s_{kj} - s_{ki}$. According to section 3.1, we aim to maximize the AP score, or equivalently minimize the following

AP risk of the query video $V_k$:

$$AP_k^{\downarrow}(f) = 1 - AP_k(f) = \frac{1}{|S^{k+}|} \sum_{s_{ki} \in S^{k+}} \frac{\sum_{s_{kj} \in S^{k-}} \mathcal{H}(d_{ji}^k)}{1 + \sum_{s_{kj} \in S^{k+} \cup S^{k-}} \mathcal{H}(d_{ji}^k)}. \tag{6}$$

This AP risk is not differentiable due to the discontinuous function $\mathcal{H}(\cdot)$. To this end, previous methods such as Smooth-AP [4] employ the Sigmoid function as a surrogate function:

$$\mathcal{G}(x; \tau) = (1 + \exp(-x/\tau))^{-1} \approx \mathcal{H}(x), \tag{7}$$

which results in an approximation risk, $i.e.$, $\widetilde{AP}_k^{\downarrow}(f)$. When $\tau \to 0$, the $\widetilde{AP}_k^{\downarrow}(f) \to AP_k^{\downarrow}(f)$, thus the approximation error of the Smooth-AP loss is small.

**Although Smooth-AP provides a straightforward solution to address the non-differentiable problem of AP, it might suffer from a gradient vanishing issue.** Specifically, as shown in fig. 4, when the score of a negative instance $s_{kj}$ significantly exceeds that of a positive instance $s_{ki}$, $i.e.$ $d_{ji}^k \gg 0$, the corresponding gradient is expected to be large such that the similarity function $f$ can be corrected. However, as depicted in fig. 4a, the gradient magnitude tends to 0, leading to slow convergence and sub-optimal solutions. This phenomenon is more evident in video similarity learning since the partial matching property (see section 3.3.1) makes $d_{ji}^k$ more likely to fall into the gradient-vanishing area.

To avoid this issue, we aim to propose a novel AP loss. To begin with, we argue that it is unnecessary to replace all $\mathcal{H}(\cdot)$. Notice that the original AP risk in eq. (6) can be reformulated as:

$$AP_k^{\downarrow}(f) = \frac{1}{|S^{k+}|} \sum_{s_{ki} \in S^{k+}} h \left( \frac{\sum_{s_{kj} \in S^{k-}} \mathcal{H}(d_{ji}^k)}{1 + \sum_{s_{kj} \in S^{k+}} \mathcal{H}(d_{ji}^k)} \right), \tag{8}$$

where $h(x) = \frac{x}{1+x}$ is a monotonically increasing function. Then, the non-differentiable terms $\mathcal{H}(d_{ji}^k)$ can be divided into two types: **1)** The positive-negative pair $(s_{kj} \in S^{k-})$ in the numerator, which should be minimized to ensure the correct ranking; **2)** The positive-positive pair $(s_{kj} \in S^{k+})$ in the denominator, which plays a role of weights. From this perspective, we only need to ensure that the surrogate loss of the former has an appropriate gradient, while for the latter we can directly use the original rankings such that the importance of each term can be precisely measured.

Motivated by the above observation, for positive-positive pairs we still utilize the Heaviside function:

$$\mathcal{R}^+(x) = \mathcal{H}(x). \tag{9}$$

As for the positive-negative pairs, the derivative of the surrogate loss should be large for the wrongly ranked pairs, $i.e.$ $d_{ji}^k + \delta \geq 0$ for a given margin $\delta > 0$. Besides, the surrogate loss should be convex such that the derivative is (non-strictly) monotonically increasing. Therefore, we design the following surrogate loss for positive-negative pairs:

$$\mathcal{R}^-(x; \delta) = \begin{cases} \mathcal{H}(-x) \cdot \frac{1}{\delta^2} x^2 + \frac{2}{\delta} x + 1, & \text{if } x \geq -\delta. \\ 0, & \text{if } x < -\delta. \end{cases} \tag{10}$$

The curves of $\mathcal{R}^-(x; \delta)$ and its derivative are visualized in fig. 4c and fig. 4d. Obviously, the above surrogate loss satisfies our design principles. Furthermore, by introducing an extra parameter $\rho$ to

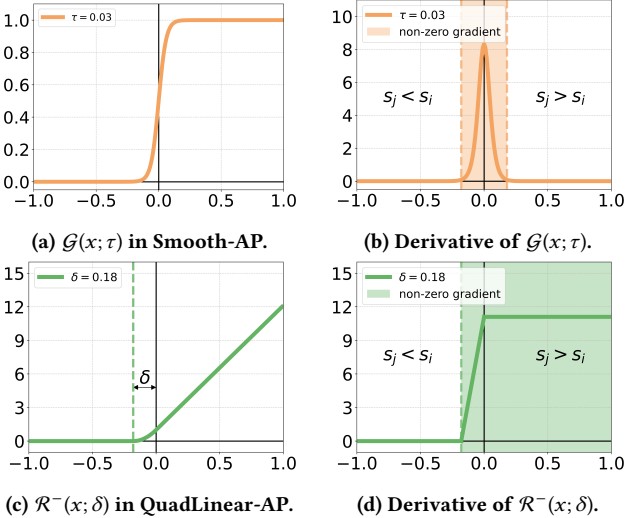

**(a)** $\mathcal{G}(x; \tau)$ **in Smooth-AP.**   **(b) Derivative of** $\mathcal{G}(x; \tau)$.

**(c)** $\mathcal{R}^-(x; \delta)$ **in QuadLinear-AP.**   **(d) Derivative of** $\mathcal{R}^-(x; \delta)$.

**Figure 4: The curves of Sigmoid function in Smooth-AP ($\tau = 0.03$) and surrogate loss function for positive-negative pairs in QuadLinear-AP ($\delta = 0.18$) and their derivative functions. The colored parts in (b) and (d) represent non-zero gradient areas of corresponding functions.**

adjust the weight of positive-positive pairs, the score distribution between positive and negative instances can be balanced well. The analysis above induces the formulation of the following AP loss, namely QuadLinear-AP:

$$\widehat{AP}_k^{\downarrow}(f) = \frac{1}{|S^{k+}|} \sum_{s_{ki} \in S^{k+}} h \left( \frac{\sum_{s_{kj} \in S^{k-}} \mathcal{R}^-(d_{ji}^k; \delta)}{1 + \rho \sum_{s_{kj} \in S^{k+}} \mathcal{R}^+(d_{ji}^k)} \right), \tag{11}$$

which enjoys the following attractive properties:

- **Differentiable AP optimization** QuadLinear-AP is differentiable for AP term, making it possible to backpropagate gradients in the learning process.
- **Suitable gradients for low AP area.** Persistent and suitable gradients in the loss function force model to correct wrongly ranked positive-negative pairs, avoiding gradient vanishing in the low AP area.
- **Favorable mathematical properties.** QuadLinear-AP is continuous, convex, and (non-strictly) monotonically increasing, ensuring a stable convergence during optimization.

As formulated in eq. (12), the final AP loss is calculated by averaging QuadLinear-AP across all query videos, which is then applied to the video-level similarity learning process. Clearly, this objective is aligned with the evaluation metric.

$$\mathcal{L}_{QLAP}^V = \frac{1}{N} \sum_{k=1}^{N} \widehat{AP}_k^{\downarrow}(f). \tag{12}$$

## 3.4 Frame Similarity Distillation

As discussed in section 1, two relevant video instances may not be completely relevant at the frame level due to the noticeable variation in the temporal dimension, $i.e.$, only several frames are highly

relevant with a query frame while the others are relatively low in actual. Therefore, solely optimizing $f$ on video-level instances proves inadequate. Next, we dive into the frame-level learning.

Given a query frame, it is hard to locate the relevant frames from another video without fine-grained annotations. A possible route is leveraging self-distillation methods [8, 56], which refines image features by distilling ensemble information from a mean teacher to the target model in a self-supervised manner. Unfortunately, since our feature extractor $g$ is trained with video data, it might ignore some image-level information. In this case, the pseudo labels generated by $g$ cannot provide more informative supervision.

Consequently, we introduce another feature extractor $g' : \mathcal{X} \mapsto \mathbb{R}^{T \times D'}$, where $D'$ is the embedding dimension. The feature extractor is pre-trained on image data with a self-supervised learning algorithm DINO [8], and the parameters are frozen. Given a video pair $V, V'$, we use $g'$ to extract features for all frames and compute the following frame-level similarities, where $V_x$ and $V'_y$ are the $x$-th frame of $V$ and $y$-th frame of $V'$, respectively.

$$S'(V, V')_{x,y} = \frac{g'(V_x)^\top g'(V'_y)}{\|g'(V_x)\|_2 \|g'(V'_y)\|_2}. \tag{13}$$

As shown in previous study [18], the similarities are highly correlated to the relevance. However, for different queries, the similarity distributions of its relevant/irrelevant frames are various, hence discretizing them into binary pseudo labels with fixed thresholds is impractical. Instead, we identify the frames with the highest/lowest similarities as positive/negative, leading to the following pseudo labels:

$$\hat{Y}_{x,y} = \begin{cases} 1, & \text{if } S'(V, V')_{x,y} \geq S'(V, V')_{x, [r_t \times T']}, \\ 0, & \text{if } S'(V, V')_{x,y} \leq S'(V, V')_{x, [(1-r_b) \times T']}, \end{cases} \tag{14}$$

where $r_t, r_b < 1$ are tunable hyperparameters, $S'(V, V')_{x,[k]}$ refers to the $k$-th largest value in $\{S'(V, V')_{x,y}\}_{y=1}^{T'}$.

Notice the varying similarity distributions across different video types, it's suboptimal to set a fixed threshold for positive or negative frames to exceed during the training phase. A feasible solution is training the model to learn to rank positive frames above the negative ones. Resembling the method in video-level learning, we optimize the frame-level similarities by $\mathcal{L}_{QLAP}^F$, which can be implemented by substituting the video instances with frame instances.

Following previous methods on ranking optimization [13, 51], we combine a basic loss $\mathcal{L}_{base}$ with the AP losses to promote collaborative optimization between ranking and representation learning. The basic loss comprises the InfoNCE loss [40] to support representation learning and an SSHN loss [32] for hard negative mining. Please refer to *supplementary material* for details.

Ultimately, the total loss for hierarchical similarity learning is formulated in eq. (15), where $\lambda_f$ and $\lambda_v$ are hyperparameters for the trade-off between components, leading to the final optimization algorithm as summarized in algorithm 1.

$$\mathcal{L} = \underbrace{\lambda_f \mathcal{L}_{QLAP}^F}_{frame-level} + \underbrace{\lambda_v \mathcal{L}_{QLAP}^V + \mathcal{L}_{base}}_{video-level} \tag{15}$$

---

**Algorithm 1** Hierarchical Average Precision Optimization

---

**Input:** Training set $S$, maximum iterations $L$, learning rate $\{\eta_l\}_{l=1}^L$, positive frame rate $r_t$, negative frame rate $r_b$.
**Output:** Model parameters $\Theta_{L+1}$.
1: Initialize model parameters $\Theta_1$.
2: **for** $l = 1$ to $L$ **do**
3:      Sample a batch of videos $\{V_i\}_{i=1}^N$ form $S$.
4:      Extract video embeddings $g(V_i)$ and $g'(V_i)$.
5:      Generate pseudo labels $\hat{Y}$ based on $r_t$ and $r_b$.
6:      Calculate similarities with function $f$ in eq. (5).
7:      Compute $\widehat{AP}_k^\downarrow(f)$ with eq. (11) to form $\mathcal{L}_{QLAP}^V$ and $\mathcal{L}_{QLAP}^F$.
8:      Compute the total loss $\mathcal{L}$ by eq. (15).
9:      Update parameters: $\Theta_{l+1} = \Theta_l - \eta_l \nabla \mathcal{L}$.
10: **end for**

---

## 4 EXPERIMENTS

In this section, we begin with a brief overview of the basic settings, including the datasets, evaluation metrics, and implementation details. Next, we compare our proposed learning framework with several previous methods on three benchmark datasets. Finally, we conduct an ablation study to evaluate the performance of different modules. For further details, please see the *supplementary material*.

### 4.1 Experimental Setup

*Datasets.* Our model is trained on the unlabeled subset of VCDB dataset[27] (we denote the core data and distractors as $C$ and $\mathcal{D}$, respectively) and evaluated on EVVE[47], SVD [26], and FIVR-5K/FIVR-200K [28]. For the FIVR dataset, we report the results of three specific subtasks: DSVR/DSVD, CSVR/CSVD, and ISVR/ISVD.

*Evaluation Metrics.* For retrieval tasks, we adopt Mean Average Precision (**mAP**) as the evaluation metric. Specifically, mAP calculates the average AP scores for each query independently and then averages these scores to reflect the model's overall ranking performance. For detection tasks, we employ Micro Average Precision (**$\mu$AP**), a metric widely used in previous studies [32, 34, 41, 42]. The $\mu$AP calculates the AP across all queries simultaneously, demonstrating the model's capability to consistently apply a uniform threshold across various queries to detect relevant instances.

*Implementation Details.* Given an input video, we generate two video clips by applying random augmentations that include temporal manipulations [29, 32], spatial transformations [12, 42], and other basic augmentations. For the feature extractor, we adopt ResNet50 [21] following [29, 32, 33], and for the pseudo label generator, we utilize DINO [8] pretrained ViT-small [14] with a patch size of 16. Our model is trained for 30,000 iterations with a batch size of 64. We use AdamW [38] with the Cosine Annealing scheduler for parameters optimization. The learning rate is set to $4 \times 10^{-5}$ with a warm-up period [37], and weight decay is set to $1 \times 10^{-2}$.

*Competitors.* We evaluate HAP-VR against various leading video retrieval methods, categorized into two types. **1) Supervised methods** include DML [31], TMK [44], TCA [52], ViSiL [29], DnS [33] with an attention student network ($S_a$) and with a binarization student network ($S_b$). **2) Unsupervised methods** include LAMV [2],

**Table 1: Comparison between video retrieval methods on EVVE, SVD, and FIVR-200K with mAP (%) of retrieval task and $\mu$AP (%) of detection task. $\dagger$ indicates the results taken from the original paper. Missing values indicate the lack of implementation or original results. The first and second best results are highlighted in soft red and soft blue, respectively.**

| Method | Label | Trainset | Retrieval (mAP) | | | | | Detection ($\mu$AP) | | | | |
| --- | --- | --- | --- | --- | --- | --- | --- | --- | --- | --- | --- | --- |
| | | | EVVE | SVD | FIVR-200K | | | EVVE | SVD | FIVR-200K | | |
| | | | | | DSVR | CSVR | ISVR | | | DSVD | CSVD | ISVD |
| DML$^\dagger$ [31] | ✓ | VCDB ($\mathcal{C}\&\mathcal{D}$) | 61.10 | 85.00 | 52.80 | 51.40 | 44.00 | 75.50 | / | 39.00 | 36.50 | 30.00 |
| TMK$^\dagger$ [44] | ✓ | internal | 61.80 | 86.30 | 52.40 | 50.70 | 42.50 | / | / | / | / | / |
| TCA [52] | ✓ | VCDB ($\mathcal{C}\&\mathcal{D}$) | 63.08 | 89.82 | 86.81 | 82.31 | 69.61 | 76.90 | 56.93 | 69.09 | 62.28 | 49.24 |
| ViSiL$^\dagger$ [29] | ✓ | VCDB ($\mathcal{C}\&\mathcal{D}$) | 65.80 | 88.10 | 89.90 | 85.40 | 72.30 | 79.10 | / | 75.80 | 69.00 | 53.00 |
| DnS ($S_a$) [33] | ✓ | DnS-100K | 65.33 | 90.20 | 92.09 | 87.54 | 74.08 | 74.56 | 72.24 | 79.66 | 69.51 | 54.20 |
| DnS ($S_b$) [33] | ✓ | DnS-100K | 64.41 | 89.12 | 90.89 | 86.28 | 72.87 | 75.80 | 66.53 | 78.05 | 68.52 | 53.48 |
| LAMV$^\dagger$ [2] | ✗ | YFCC100M | 62.00 | 88.00 | 61.90 | 58.70 | 47.90 | 80.60 | / | 55.40 | 50.00 | 38.80 |
| VRL$^\dagger$ [23] | ✗ | internal | / | / | 90.00 | 85.80 | 70.90 | / | / | / | / | / |
| ViSiL$_f$$^\dagger$ [29] | ✗ | ImageNet | 62.70 | / | 89.00 | 84.80 | 72.10 | 74.60 | / | 66.90 | 59.50 | 45.90 |
| S$^2$VS [32] | ✗ | VCDB ($\mathcal{D}$) | 67.17 | 88.40 | 92.53 | 87.73 | 74.51 | 80.72 | 65.04 | 86.12 | 77.41 | 63.26 |
| HAP-VR (Ours) | ✗ | VCDB ($\mathcal{D}$) | 69.15 | 89.00 | 92.83 | 88.21 | 74.72 | 82.88 | 67.87 | 88.41 | 79.85 | 64.79 |

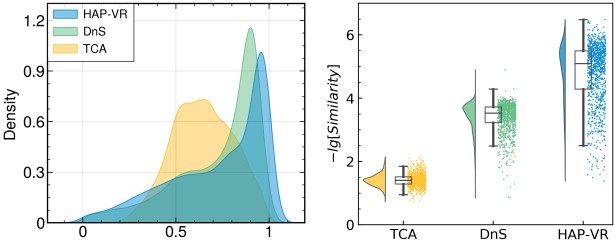

(a) Relevant pair distribution. (b) Irrelevant pair distribution.

**Figure 5: Similarity distribution of relevant and irrelevant instance pairs for HAP-VR, DnS, and TCA on the DSVD set of FIVR-200K. All similarities are rescaled to $[0, 1]$.**

VRL [23], ViSiL$_f$ [29] (baseline of ViSiL without training), and S$^2$VS [32].

## 4.2 Evaluation Results

The overall performance on video retrieval and detection tasks above is reported in table 1, leading to several key conclusions: **1)** HAP-VR stands out among other unsupervised or self-supervised methods in both mAP and $\mu$AP metrics, with an average improvement of **0.71%** and **2.25%**, respectively. These outcomes underscore the effectiveness of aligning the training objectives with the evaluation metrics, directly enhancing the average precision. **2)** Detection tasks enjoy larger performance gains than retrieval tasks. This is primarily due to the more pronounced imbalance between instances in detection tasks. By emphasizing the overall rankings of the positive instances, HAP-VR achieves a more optimal similarity distribution across all queries, resulting in a notable increase in $\mu$AP. **3)** Compared with supervised methods, HAP-VR achieves a better overall performance. To investigate the underlying reason, we visualize

the video similarity distributions in fig. 5. Compared with the supervised methods, HAP-VR establishes a clearer margin between scores of relevant and irrelevant pairs. Since annotations are based on the video categories, the supervised model tends to distinguish the pre-defined categories but not video instances. Accordingly, when encountering videos beyond these pre-defined categories, the model is prone to overfit the categories, which hinders discriminating between negative instances, thereby reducing the model's transferability.

## 4.3 Ablation Study

*Ablation results on proposed QuadLinear-AP loss.* To validate the effectiveness of the proposed QuadLinear-AP loss, we make a comparison with other commonly used losses, which can be categorized into three types: **1) Point-wise losses**, include Mean Absolute Error (MAE) and Mean Squared Error (MSE). These losses measure the discrepancy between predicted scores and actual labels for each item independently. **2) Pair-wise losses**, include Contrastive loss [17], Triplet loss [49] and Circle loss [54]. These losses focus on distinguishing between the positive and negative instances in pairs. **3) List-wise losses**, include FastAP [6], DIR [46], BlackBox [43], and Smooth-AP [4]. These approaches optimize the model directly based on ranking metrics such as AP.

For a straightforward comparison, we only combine these losses with $\mathcal{L}_{base}$ and train the models using 10% of the VCDB ($\mathcal{D}$) for 10,000 iterations. Except for the specific hyperparameters associated with each loss, all other settings remain constant to ensure a fair comparison.

The comparison results are presented in table 2. From these results, we can draw the following conclusions: **1)** In general, list-wise losses outperform point-wise and pair-wise losses, supporting our motivation to develop an AP-oriented method for video retrieval tasks. **2)** QuadLinear-AP achieves an average improvement of about

**Table 2: Comparison between QuadLinear-AP and other loss functions on the FIVR-5K with mAP (%) of retrieval task and $\mu$AP (%) of detection task. The first and second best results are highlighted in soft red and soft blue, respectively.**

| Losses | Retrieval (mAP) | | | Detection ($\mu$AP) | | |
|---|---|---|---|---|---|---|
| | DSVR | CSVR | ISVR | DSVD | CSVD | ISVD |
| MAE | 89.07 | 88.03 | 80.86 | 78.08 | 75.69 | 65.26 |
| MSE | 89.22 | 88.26 | 80.80 | 78.66 | 76.07 | 65.44 |
| Contrastive [17] | 88.67 | 88.09 | 80.97 | 75.12 | 74.23 | 67.41 |
| Triplet [49] | 88.11 | 87.77 | 81.21 | 72.94 | 73.18 | 69.23 |
| Circle [54] | 87.53 | 86.11 | 78.77 | 73.26 | 71.15 | 59.33 |
| FastAP [6] | 89.30 | 88.42 | 81.16 | 78.83 | 77.51 | 69.95 |
| DIR [46] | 89.65 | 88.57 | 80.64 | 78.50 | 76.22 | 65.42 |
| BlackBox [43] | 89.70 | 88.55 | 80.53 | 80.07 | 77.37 | 66.00 |
| Smooth-AP [4] | 89.36 | 88.33 | 80.73 | 79.85 | 77.75 | 68.42 |
| QuadLinear-AP (**Ours**) | 90.80 | 89.68 | 81.31 | 82.92 | 80.03 | 71.45 |

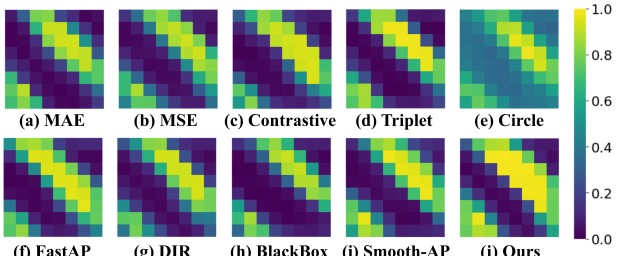

(a) MAE  (b) MSE  (c) Contrastive  (d) Triplet  (e) Circle

(f) FastAP  (g) DIR  (h) BlackBox  (i) Smooth-AP  (j) Ours

**Figure 6: Heatmaps of similarity matrices generated by various losses. In contrast, our QuadLinear-AP distinguishes between relevant and irrelevant instances more clearly.**

**1.84%** on mAP and **2.87%** on $\mu$AP over other list-wise losses, reflecting the effectiveness of the proposed AP loss. The visualization of frame-level similarity shown in fig. 6 illustrates that QuadLinear-AP presents a clearer distinction between relevant and irrelevant instances compared to other competitors.

*Ablation results on proposed modules.* Comparing Line 1 with Line 2 in table 3, the application of the TopK-Chamfer Similarity measure yields average boosts of **0.59%** on mAP and **1.13%** on $\mu$AP based on the baseline model. This suggests the efficacy of the TopK-Chamfer Similarity measure, which will be further discussed in the ablation study on similarity measures. Comparing Line 2 with Line 3 shows that incorporating video-level AP optimization further enhances performance in retrieval and detection tasks, with increases of **0.69%** and **2.13%** respectively. Such improvements reveal the necessity of aligning training objectives with evaluation metrics. Moreover, implementing a frame-level learning process further improves the overall outcomes, emphasizing the value of learning the internal similarity within the video precisely.

*Ablation results on similarity measure.* To validate the effectiveness of the proposed TopK-Chamfer Similarity measure, we evaluate the model performances varying the top-k rate $k_t$. Note that when $k_t = 0.0$, the measure can be seen as the original Chamfer Similarity;

**Table 3: Results in the ablation study of modules including TopK-Chamfer Similarity measure, video-level AP loss, and frame-level AP loss. Improvements in performance compared to the baseline are denoted with red subscripts.**

| $\mathcal{L}_{base}$ | TopK. Sim. | $\mathcal{L}_{QLAP}^{V}$ | $\mathcal{L}_{QLAP}^{F}$ | EVVE | FIVR-5K | | |
|---|---|---|---|---|---|---|---|
| | | | | | DSVR/DSVD | CSVR/CSVD | ISVR/ISVD |
| **Retrieval (mAP)** | | | | | | | |
| ✓ | | | | 67.64 | 88.18 | 87.16 | 80.14 |
| ✓ | ✓ | | | $69.41_{+1.77}$ | $88.36_{+0.18}$ | $87.42_{+0.26}$ | $80.30_{+0.16}$ |
| ✓ | ✓ | ✓ | | $69.55_{+1.91}$ | $89.39_{+1.21}$ | $88.54_{+1.38}$ | $80.79_{+0.65}$ |
| ✓ | ✓ | ✓ | ✓ | $69.58_{+1.94}$ | $89.75_{+1.57}$ | $88.59_{+1.43}$ | $80.72_{+0.58}$ |
| **Detection ($\mu$AP)** | | | | | | | |
| ✓ | | | | 79.13 | 75.49 | 73.84 | 63.50 |
| ✓ | ✓ | | | $80.67_{+1.54}$ | $76.23_{+0.74}$ | $74.77_{+0.93}$ | $64.81_{+1.31}$ |
| ✓ | ✓ | ✓ | | $81.09_{+1.96}$ | $78.64_{+3.15}$ | $77.04_{+3.20}$ | $68.23_{+4.73}$ |
| ✓ | ✓ | ✓ | ✓ | $82.96_{+3.83}$ | $81.56_{+6.07}$ | $78.32_{+4.48}$ | $66.87_{+3.37}$ |

**Table 4: Results in the ablation study of similarity measure. In particular, * represents using Chamfer Similarity and † represents using average pooling. The first and second best results are highlighted in soft red and soft blue, respectively.**

| $k_t$ | Retrieval (mAP) | | | | Detection ($\mu$AP) | | | |
|---|---|---|---|---|---|---|---|---|
| | EVVE | FIVR-5K | | | EVVE | FIVR-5K | | |
| | | DSVR | CSVR | ISVR | | DSVD | CSVD | ISVD |
| 0.00* | 67.57 | 89.52 | 88.38 | 80.55 | 78.85 | 78.78 | 76.93 | 65.99 |
| 0.03 | 68.98 | 89.65 | 88.69 | 80.91 | 80.70 | 79.01 | 77.01 | 68.21 |
| 0.06 | 69.55 | 89.39 | 88.54 | 80.79 | 81.09 | 78.64 | 77.04 | 68.23 |
| 0.10 | 69.03 | 87.87 | 87.12 | 79.69 | 80.75 | 73.19 | 71.75 | 61.96 |
| 0.20 | 68.69 | 85.26 | 85.01 | 78.01 | 81.54 | 69.12 | 68.35 | 59.57 |
| 0.30 | 68.27 | 81.54 | 81.98 | 75.93 | 80.04 | 64.30 | 65.41 | 58.41 |
| 1.00† | 55.49 | 61.29 | 64.25 | 62.84 | 77.32 | 37.57 | 44.27 | 42.92 |

when $k_t = 1.0$, the measure is equal to average pooling. As indicated by the results in table 3, the optimal performance is achieved neither at $k_t = 0.0$ nor at $k_t = 1.0$. This outcome supports the utility of selecting top-K values. From another perspective, the best performance is obtained when $k_t$ is small, demonstrating the capability of the TopK-Chamfer Similarity in diminishing redundancy and reducing the influence of noise, thereby ensuring robustness in similarity calculation.

## 5  CONCLUSION

In this paper, we design a self-supervised framework for video retrieval, which features a video-oriented similarity measure to gather fine-grained features and a novel AP-based loss with reasonable gradients to correct mis-ranked instance pairs efficiently, filling the gap between the training objective and evaluation metric. Within the framework, we propose a hierarchical learning strategy to conduct AP optimization both on video and frame levels, which generates precise estimations of the AP loss, thus enhancing the accuracy of the similarity learning process. Experimental results demonstrate that our framework often surpasses previous works in several benchmark datasets, making it a feasible solution for video retrieval tasks. In future work, we plan to extend our framework to other applications, which we hope could support subsequent research to further contribute to the multimedia community.

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
