# OpenReview forum: "Not All Pairs are Equal: Hierarchical Learning for Average-Precision-Oriented Video Retrieval"
_acmmm.org/ACMMM/2024/Conference — MM2024 Oral_

### Official Review · Reviewer_kjjn · 2024-05-16

**Rating:** 5
**Confidence:** 3

**Summary:**

The paper presents a fine-grained video retrieval approach with an average-precision-based loss function.

**Strengths:**

The paper is generally well-written and describes the method in detail. While it is somewhat math-heavy, one can see that the authors tried not to complicate matters needlessly. The supplementary material contains further details and even the code of the implementation. While the authors do not explicitly mention it, this leads one to hope that they intend to publish their implementation as open-source software.

**Limitations:**

The paper has some minor linguistic and formatting issues. Specifically, it neglects to capitalize references to figures and sections. For a final version, it should undergo some additional proofreading.

**Suitability:**

2

---

### Official Review · Reviewer_RsV7 · 2024-05-22

**Rating:** 6
**Confidence:** 3

**Summary:**

This paper proposes to tackle the gap between the training and  evaluation stages in the domain of video retrieval.
They propose a new methodology HAP-VR with a dedicated hierarchical similarity metric and learning framework.
Exhaustive experiments are presented to validate the proposal.

**Strengths:**

Major strength.
This paper is very well-written and easy to read. The authors clearly articulate their challenges with pedagogical figures, which is commendable.

Each step of their framework is justified by a thorough analysis of state-of-the-art references.

This paper makes a highly interesting contribution to the scientific community.

Exhaustive experiments and their performance evaluations are also among the strengths of this paper.

**Limitations:**

Minor limitations.
The paper is clearly structured and written similarly to the S2VS paper; however, its originality is distinctly different. The comparison result table evaluates the proposed method against other recent methodologies, with half of them developed by the same researchers. The other half of the comparisons include several instances of missing performance data due to unavailability - which is regretful.

At the end, a small typo, Table 3 should be Table 4. And all figures and tables should start with a capital letter.

I think the ablation study analysis if limited to the content of the article is not detailed enough to explain clearly the given tables and figures.

However, the supplementary material file completes greatly this article.

**Suitability:**

3

---

### Official Review · Reviewer_wbpW · 2024-05-23

**Rating:** 5
**Confidence:** 2

**Summary:**

The paper introduces several improvements in the methodology for video retrieval. The three main contributions include a hierarchical video retrieval architecture, a surrogate loss function that approximates AP loss with better differentiability and a different similarity measure. The authors provide theoretical grounding for their contributions and evaluate against a number of competitor systems.

**Strengths:**

The theoretical justifications provide a strong argument for their methodology. In particular: vanishing gradients can be problematic and the derivative of their QuadLinear-AP is more favorable to optimization than the Smooth-AP they compare themselves to. Their proposed TopK-Chamfer similarity can provide more fine-grained information since it focuses on the most relevant instances. The proposed architecture seems theoretically sound and is explained well.

All claims in the paper are supported through both theory and experiments. The source is code is provided in a well-organized folder. At a glance the pytorch implementation appears reasonable. Both theory and results are further supported by supplementary material, keeping the main paper concise and providing additional insight.

**Limitations:**

It seems like the paper is an incremental improvement over previous methods. Many of the reference systems are developed by Giorgos Kordopatis-Zilos and do not seem to be state of the art on most video-retrieval datasets. His systems do show strong results on the FIVR dataset, which was created by the same author. There are comparisons to other unrelated systems, it just seems to be biased toward this particular researcher’s work. That said, there are comparisons on the EVVE and SVD datasets as well and this might not be a problem.

**Suitability:**

3

---

### Official Review · Reviewer_LA4m · 2024-05-24

**Rating:** 5
**Confidence:** 3

**Summary:**

The authors introduce a novel method for improving video retrieval based on the self-supervised Hierarchical learning framework considering the Average precision during training. To this end, they aim to tackle the challenges for calculating the similarity by taking into account all the frames/features of the videos and addressing the vanish gradient problem that emerged in the cases where the samples are strongly mis-ranked. Specifically, they propose a novel loss for AP named, TopK-Chamfer Similarity and QuadLinear-AP loss to measure and optimize video-level similarities, alongside constraining frame-level similarities to produce a precise estimation of AP loss.

**Strengths:**

The paper's strengths are the novelty that is fully described and explained in the introduction section.  Specifically, the authors, present the significance of the video retrieval systems and the way these systems are evaluated (Average precision), and then they clearly illustrate their contributions based on a metric-learning-based approach. The work is comprehensibly presented and illustrated. The authors provide adequate state of the art of the related methods. They explain the state-of-the-art focus on fine-grained methods and detail illustrate the methods related to video retrieval as the main objective of the proposed work. In the same section, they also report the state of the art of the methods aiming to optimize metrics for improving their retrieval precision. In the methodology section the authors, provide a detailed explanation of the problem, making clear to the reader what are the objectives of the proposed work. They prove and explain both textually and mathematically, which helps the reader follow such a complex pipeline. Finally, the experimental setup section includes all the experiments to establish a fair comparison of the proposed method compared to the state-of-the-art.  The authors followed a common approach to prove that the proposed method outperforms the state-of-the-art, based on retrieval and detection evaluation. They also provide a necessary ablation study part that covers the unevaluated aspects during the main evaluation.

**Limitations:**

The proposed work reports a novel approach followed by a detailed technical justification. Although the work is well-written and comprehensively illustrated there are lacks in a few points. Please take care of the reported comments to improve the presentation of the work.
•	In line 12, “for the users” and in line 43, please remove “which”.
•	In Figure 2, there is no explanation of the TR and TQ axis, both in the text and in the caption.
•	In lines 141 to 143, more details and an explanation of the phrase “fall into the gradient vanishing..” e.g. when aiming to maximize AP or minimize 1-AP similar to Smooth-AP losses.
•	In line 155 the word “proper” gradients need further clarification.
•	Section 2.1.2 consists of a very long paragraph that should be split into two paragraphs.
•	In lines 322-327, there is a different formation of illustrating the methods. Update the formation of illustrating the methods to improve consistency. It would be better to keep reporting on the method names/cross-references and not provide author names.
•	Figure 3, the colour of the input batch sampling does not follow a consistent sequence in both streams, this should be explained in the text why inputting batches in a sequence of orange, red, blue, and green we extract embeddings of orange, red, green, and blue.
•	In line 362, please check the reference of the Heaviside function if it is the most representative of the definition.
•	In lines 368-369, there is confusion about what is illustrated in Fig 3 and whether there are input two videos or batches. Please rephrase and make it more clear either to the text or to the figure.
•	Lines 395-397, need further clarification when using the term of frame-level similarity and the term patch-level similarity.
•	In line 402 please update similarly to the above comment related to the author's name.
•	In line 467, this is the justification related to the vanish of gradient, minimizing the 1-AP. This is related to the comment for lines In lines 141 to 143.
•	In line 482, there is no specific reference to a specific tile of Fig 4, e.g. (a). Moreover, all the tiles belonging to Fig 4 should also be explained in the text.

**Suitability:**

3

---

### Meta-Review · Area_Chair_HBH3 · 2024-07-02

**Recommendation:** Accept (Oral)
**Confidence:** 4

**Metareview:**

This paper proposes a framework for assessing video retrieval results by considering the similarity of results, and therefore an alternative to the typical average precision metric. As confirmed by all reviewers, this is a novel approach and the paper not only provides a good motivation and is written well and clear, but also the claims of the work are confirmed by a sound and exhaustive evaluation with good results. Moreover, the authors provide the source code of their implementation. Overall, this is an interesting and novel contribution to the multimedia community.